Characterization of four new monoclonal antibodies against the distal N-terminal region of PrPc

Didonna Alessandro 1 *
Venturini Anja Colja 2
Hartman Katrina 2
Vranac Tanja 2
Čurin Šerbec Vladka 2
Legname Giuseppe 1 3 giuseppe.legname@sissa.it
1 Department of Neuroscience, Scuola Internazionale Superiore di Studi Avanzati (SISSA) , Trieste , Italy
2 Department for Production of Diagnostic Reagents and Research, Blood Transfusion Centre of Slovenia , Ljubljana , Slovenia
3 ELETTRA—Sincrotrone Trieste S.C.p.A , Trieste , Italy
Grandori Rita
* Current affiliation: Department of Neurology, University of California San Francisco, San Francisco, CA, USA

Electronic publication date: 2015 Mar 19
Publication date: 2015
Volume: 3
Electronic Location ID: e811
Received 2014 Dec 14; Accepted 2015 Feb 11
Copyright: © 2015 Didonna et al.
Copyright year: 2015
Copyright holder: Didonna et al.
License: This is an open access article distributed under the terms of the Creative Commons Attribution License, which permits unrestricted use, distribution, reproduction and adaptation in any medium and for any purpose provided that it is properly attributed. For attribution, the original author(s), title, publication source (PeerJ) and either DOI or URL of the article must be cited.
License URL: https://creativecommons.org/licenses/by/4.0/

Keywords: Prion protein, Monoclonal antibodies, Histopathology, Prion inhibition

Funding: European Community’s Seventh Framework Programme FP7/2007–2013 222887 Slovenian Research Agency with Research Programme P4-0176 Research Project L3-0206 PhD grant This study was supported by the European Community’s Seventh Framework Programme (FP7/2007–2013) under grant agreement no 222887—the PRIORITY project to Giuseppe Legname, and by the Slovenian Research Agency with Research Programme P4-0176, Research Project L3-0206 and with a PhD grant to Anja Colja Venturini. The funders had no role in study design, data collection and analysis, decision to publish, or preparation of the manuscript.

==============================
Prion diseases are a group of fatal neurodegenerative disorders that affect humans and animals. They are characterized by the accumulation in the central nervous system of a pathological form of the host-encoded prion protein (PrPC). The prion protein is a membrane glycoprotein that consists of two domains: a globular, structured C-terminus and an unstructured N-terminus. The N-terminal part of the protein is involved in different functions in both health and disease. In the present work we discuss the production and biochemical characterization of a panel of four monoclonal antibodies (mAbs) against the distal N-terminus of PrPC using a well-established methodology based on the immunization of Prnp0/0 mice. Additionally, we show their ability to block prion (PrPSc) replication at nanomolar concentrations in a cell culture model of prion infection. These mAbs represent a promising tool for prion diagnostics and for studying the physiological role of the N-terminal domain of PrPC.

Introduction

Transmissible spongiform encephalopathies (TSEs) are a group of fatal neurodegenerative diseases that occur in human and animals. They can be sporadic, inherited and iatrogenic (Prusiner, 1988) and include Creutzfeldt-Jakob disease (CJD), fatal familial insomnia (FFI) and Gerstmann–Straüssler–Scheinker syndrome (GSS) in humans, bovine spongiform encephalopathy (BSE) in cattle, scrapie in sheep and chronic wasting disease (CWD) in deer, moose and elk.

The unique agent responsible for these maladies is a pathological conformer (PrPSc) of the host-encoded prion protein (PrPC). Upon conversion, most α-helix motives are replaced by β-sheet secondary structures (Kuwata et al., 2002; Pan et al., 1993). This event changes dramatically the biochemical properties of PrPC, which becomes partially resistant to proteases, detergent-insoluble and prone to aggregation (Cohen & Prusiner, 1998).

PrPC is a ubiquitous glycoprotein expressed mainly in the central nervous system (CNS). It is linked to the cell membrane via a glycosylphosphatidylinositol (GPI) anchor, and localized within cholesterol-rich domains called rafts. The physiological role of PrPC is still enigmatic; PrPC-null mice failed to show any gross phenotypic feature (Raeber et al., 1998) and no univocal role has been proposed yet (Didonna, 2013).

PrPC consists of two domains: the globular C-terminus of the protein contains three α-helices and an anti-parallel β-sheet, while the evolutionarily highly conserved N-terminus is flexible and mostly unstructured (Zahn et al., 2000). Despite the lack of ordered structure, many lines of evidence suggest a central role of the N-terminal domain in PrPC function.

Indeed, the N-terminal part is associated with PrPC internalization (Nunziante, Gilch & Schatzl, 2003) for which the initial polybasic region (aa23–aa28 NH2-KKRPKP) was shown to be especially important (Sunyach et al., 2003). The N-terminal domain (aa23–aa90) also acts as a raft-targeting signal, as it is sufficient to confer raft localization when fused to a non-raft transmembrane-anchored protein (Walmsley, Zeng & Hooper, 2003). A recent model suggests that the N-terminus might penetrate the lipid bilayer of the plasma membrane via its polybasic regions and mediate signal transduction within the cytosol (Iraci et al., in press).

The N-terminus furthermore binds copper ions through four octapeptide repeats (PHGG(G/S)WGQ; residues aa59–aa90) and its involvement in copper endocytosis and metabolism has been demonstrated (Brown et al., 1997). Moreover, copper binding seems to promote PrPC internalization in clathrin-coated pits (Hooper, Taylor & Watt, 2008). More recently, PrPC has been shown to bind Aβ oligomers with high affinity—possibly mediating their neurotoxic effects—and the polybasic stretch at the extreme N-terminus is one of the two critical regions for the interaction (Chen, Yadav & Surewicz, 2010; Lauren et al., 2009).

Insertions and point mutations in N-terminus impair cell response to oxidative stress, implying that this domain is also required to regulate such cellular activity (Yin et al., 2006; Zeng et al., 2003). Furthermore, N-terminus mediates neuroprotection both in vitro and in vivo (Didonna et al., 2012; Flechsig et al., 2003). Additionally, a recent study has shown that PrPC flexible tail regulates the toxicity of globular domain ligands (Sonati et al., 2013).

The unstructured domain seems to participate in PrPSc formation as well. The N-terminus has been shown to influence the aggregation of PrP in vitro by promoting high-order assembled structures (Frankenfield, Powers & Kelly, 2005). For instance, the N-terminus has been recently found essential for the assembly of a specific β-sheet-rich oligomer, containing ∼12 PrP molecules (Trevitt et al., 2014). Removing the N-terminus decreased the prion conversion efficiency in vivo as well (Supattapone et al., 2001). Several inheritable forms of prion diseases are caused by mutations within this region. An increased number of octapeptides correlate with early forms of familial CJDs (Vital et al., 1999) and are shown to increase the rate of protease-resistant PrP formation (Moore et al., 2006). In addition, the polybasic region aa23–aa30 seems crucial for the correct folding of PrPC and it might regulate the acquisition of strain-specific conformations in disease (Ostapchenko et al., 2008). Another set of data highlighted the role of the N-terminus in dominant negative inhibition of prion formation. N-terminally truncated PrP(Q218K) molecules showed a reduced dominant-negative action compared to full-length forms; the authors propose a model in which the N-terminus domain stabilizes the C-terminus of the molecule (Zulianello et al., 2000).

Considering the relevance of the N-terminal domain for the physiopathology of prion protein, we have generated four monoclonal antibodies that recognize epitopes situated in the distal region of the N-terminus. In this study we present their production and exhaustive characterization, both biochemical and histopathological. A possible use as prion replication inhibitors is also described.

Materials and Methods

Ethics statement

All experiments involving animals were performed in accordance with European regulations [European Community Council Directive, November 24, 1986 (86/609/EEC)]. Experimental procedures were notified to and approved by the Italian Ministry of Health, Directorate General for Animal Health (notification of 17 Sept. 2012). All experiments were approved by the local authority veterinary service and by SISSA Ethics Committee. All reasonable efforts were made to ameliorate suffering. All mice were obtained from the European Mutant Mouse Archive.

Approval for research involving human material has been obtained from the Slovenian National Medical Ethics Committee with decision dated January 15, 2008. Post mortem brain tissue of a patient who was clinically suspected for CJD was analyzed by immunohistochemistry without patient’s consent because such analysis is obligatory by a ministerial decree in purpose of TSE surveillance (Official Gazette of the Republic of Slovenia, 2/2001). Human brain samples for immunohistochemistry were obtained from the Institute of Pathology, Faculty of Medicine, University of Ljubljana, Slovenia.

Cell lines and cell culture

GT1-1 cells and ScGT1-1 cells (kindly provided by Dr. P Mellon, The Salk Institute, La Jolla, CA, USA) were maintained in Dulbecco’s Modified Eagle’s Medium with 4.5 g/L glucose (DMEM) (GIBCO/Invitrogen, Irvine, California, USA) supplemented with 10% v/v fetal bovine serum (FBS) (GIBCO/Invitrogen, Irvine, California, USA) and antibiotics (100 IU/mL penicillin and 100 µg/mL streptomycin) (GIBCO/Invitrogen, Irvine, California, USA) at 37 °C in a humidified atmosphere with 5% CO2.

The NS1 murine myeloma cell line and all hybridoma cell lines prepared in this study were maintained in DMEM (ICN Biomedical, Carlsbad, California, USA) supplemented with 13% v/v bovine serum (HyClone, Little Chalfont, UK), 2 mM L-glutamine (Sigma, St. Louis, Missouri, USA), 130 µg/mL streptomycin (Sigma, St. Louis, Missouri, USA) and 100 IU/mL penicillin (Sigma, St. Louis, Missouri, USA).

Mouse immunization and cell fusion

Three female Prnp0/0 mice (6–8 weeks old, mixed C57BL × 129/Sv background) were immunized with full-length (aa23–aa231) oxidized recombinant human prion protein (recHuPrP), with M on codon 129 (Prionics, Zurich, Switzerland). Each mouse was immunized subcutaneously with 20 µg of the antigen in Complete Freund’s Adjuvant (final volume 200 µL) and then twice in four weeks’ intervals intraperitoneally with 20 µg of the antigen in Incomplete Freund’s Adjuvant (final volume 200 µL). Mice were bled from the tail vein and the immune sera were collected. They were tested by indirect ELISA and the most responsive animal was given a final booster with 20 µg of the antigen in physiological saline (final volume 100 µL), administered intravenously in the tail vein three days prior to cell fusion. Splenocytes were isolated and fused with mouse NS-1 myeloma cells with 50% polyethylene glycol, according to standard procedures, used in our laboratory. Cell suspension was distributed to 96-well microtiter plates and cultured in CO2 incubator at 37 °C. Hybridoma cells were grown by maintaining the cells for ten days in selective HAT medium and another week in HT medium (DMEM supplemented with hypoxanthine–aminopterin–thymidine or hypoxanthine–thymidine, respectively). The presence of specific antibodies was screened in supernatants during and after 10–14 days by indirect ELISA.

Indirect enzyme-linked immunosorbent assay (ELISA)

Indirect ELISA was performed in 96-well Nunc MaxiSorp microtiter plates (eBioscience, San Diego, California, USA). Wells were coated with 0.5 µg/mL of recHuPrP in 50 mM carbonate/bicarbonate buffer, pH 9.6, and incubated overnight at 4 °C. The next day, the plates were washed three times with washing buffer (sodium phosphate buffer, containing 150 mM NaCl, 0.05% Tween 20, pH 7.2–7.4) and blocked for 30 min at 37 °C with blocking buffer (1% BSA in washing buffer). After three washings, the plates were incubated with immune sera, serially diluted 1:10, starting dilution 1:100, for 1.5 h at 37 °C. Plates were washed again and then incubated with secondary goat anti-mouse IgG + IgM antibodies, conjugated with horseradish peroxidase (HRP) (Jackson Immunoresearch, West Grove, Pennsylvania, USA), diluted 1:5,000 in blocking buffer, for 1.5 h at 37 °C. After washing, substrate 2,2′-azino-bis(3-ethylbenzothiazoline-6-sulfonic acid) (ABTS) (Sigma, St. Louis, Missouri, USA) in citrate-phosphate buffer, pH 4.5, was added and incubated for 20 min at 37 °C. The color reaction was measured spectrophotometrically at 405 nm with a microtiter plate reader.

Hybridoma cell lines selection

Production of antibodies was monitored and tested for their specificity to recHuPrP with indirect ELISA. Selected hybridomas were cultured in DMEM until stable cell lines were established and then subcloned by limiting dilution. Finally, four monoclonal antibodies were isolated, appropriate cell lines were cultured and then frozen in liquid nitrogen for further use.

After culturing cell lines in larger volumes, the supernatants were harvested and monoclonal antibodies purified by fast protein liquid chromatography (AKTA FPLC; GE Healthcare, Little Chalfont, UK).

Immunoglobulin class and subclass were determined by indirect ELISA using anti-Fc specific antibodies.

Epitope mapping

Epitopes were analyzed by PEPSCAN (Lelystad, Netherlands), using overlapping 20-mer synthetic peptides from HuPrP (aa23–aa230), shifted by four amino acids.

Afterwards, proposed epitopes were refined with additional overlapping 12-mer synthetic peptides, from N-terminal domain (aa23–aa64 from human PrP and aa44–aa64 from mouse PrP), shifted by three amino acids. For this purpose, peptides were coated to the microtiter plate separately, at concentration 2 µg/mL, incubated with mAbs DE10, DC2, EB8 and EF2 at concentration 5 µg/mL and then incubated with secondary goat anti-mouse IgG + IgM antibodies, conjugated with HRP (Jackson Immunoresearch, West Grove, Pennsylvania, USA). Inhibition assays were performed with the same peptides. RecHuPrP was coated to the microtiter plates. Monoclonal antibodies were mixed with 100 times redundant molar peptide concentration. Mixtures were added to microtiter plates and incubated. After washing, plates were incubated with secondary, goat anti-mouse IgG + IgM antibodies, conjugated with HRP (Jackson Immunoresearch, West Grove, Pennsylvania, USA). The percentage of inhibition was calculated.

Western blot analysis

Different brain homogenates (10% w/v) were prepared from human, bovine, hamster, sheep, deer, rabbit and rat and mouse (BALB/c and Prnp0/0) brain tissues in ice-cold buffer (0.5% Nonidet P40, 0.5% Na-deoxycholate in PBS) with HT1000 Potter homogenizer. Aliquots were stored at −80 °C and centrifuged prior to use (5 min at 5,000 × g). Samples were loaded on 12% polyacrylamide gels and SDS-PAGE was performed. Proteins were blotted on 0.2 µm nitrocellulose membranes (Bio-Rad) at 200 mA for 90 min. Membranes were then blocked with 5% (w/v) non-fat milk in Tris-buffered saline/0.05% Tween-20 (TBS-T) at 4 °C overnight, washed in TBS-T and incubated with the four monoclonal antibodies (5 µg/mL in 1% non-fat milk/TBS-T) for 60 min by shaking at room temperature. Membranes were washed again and incubated for 60 min with secondary, goat anti-mouse antibodies, conjugated with HRP (Jackson Immunoresearch, West Grove, Pennsylvania, USA), in 1% non-fat milk/TBS-T, at room temperature (RT). Chemiluminescence was detected by ECL kit (Amersham, Buckinghamshire, UK). Films were exposed for 10 min.

Immunohistochemistry (IHC)

Sections of paraformaldehyde-fixed, paraffin-embedded human cerebellar tissue samples from a patient with diagnosed sporadic CJD (sCJD) with primitive plaques and synaptic prion deposition pattern were used in the study. Tissue samples were immersed in 96% formic acid for 1 h after fixing in paraformaldehyde. Sections were deparaffinized and pretreated for optimal antigen retrieval by 30 min autoclaving at 121 °C in distilled water, followed by a 5 min incubation in 96% formic acid. The sections were then blocked in 1% BSA solution for 20 min at RT. They were subsequently incubated overnight at RT in a moist chamber with all N-terminal mAbs tested at the concentration of 5 µg/mL. All sections were then washed and incubated for 1.5 h with anti-mouse HRP-labeled antibodies diluted 1:1000 (Jackson ImmunoResearch, West Grove, Pennsylvania, USA) at RT. After thorough rinsing, the sections were developed in DAB chromogen (Sigma, St. Louis, Missouri, USA) for 5 min. Brain tissue counterstaining was obtained by immersion of sections in Mayer’s hematoxylin for 2 min.

Proteinase K digestion assay

Cells were washed twice with cold PBS 1X (GIBCO/Invitrogen, Irvine, California, USA) and lysed with lysis buffer (10 mM Tris–HCl pH 8.0, 150 mM NaCl, 0.5% nonidet P-40 substitute, 0.5% deoxycholic acid sodium salt) and pelleted by centrifugation at 2,300 × g for 5 min. The supernatant was collected and the total protein concentration measured using Bicinchoninic acid assay (Pierce). For the assay, 250 µg of proteins were treated with 5 µg of proteinase K (Roche; ratio protein:protease 50:1) for 1 h at 37 °C. Digestion was stopped by adding phenylmethyl sulphonyl fluoride (PMSF) (Sigma, St. Louis, Missouri, USA) to a final concentration of 2 mM. PrPSc was precipitated by ultracentrifugation at 100,000 × g (Optima TL; Beckman Coulter, Brea, California, USA) for 1 h at 4 °C. After centrifugation, the supernatant was discarded and the pellet resuspended in loading buffer before loading onto a 12% SDS-PAGE. Twenty-five µg of total, undigested proteins were loaded as a control. Samples were electroblotted onto membranes of polyvinylidene fluoride (PVDF) (EMD Millipore, Billerica, Massachusetts, USA). After blocking in 5% non-fat milk/TBS-T for 1 h at RT, membranes were incubated in 1 µg/mL of Fab D18 (InPro Biotechnology, Inc., South San Francisco, California, USA) in PBS for 2 h at RT, followed by incubation for 1 h, in the secondary antibody goat-anti-human HRP-conjugated, (Pierce; Thermo Scientific, Waltham, Massachusetts, USA), diluted 1:5000 in 5% non-fat milk/TBS-T. After several washings the signal was detected using ECL kit (Amersham, Buckinghamshire, UK) on ECL Hypermax films (Amersham, Buckinghamshire, UK).

ERK1/2 phosphorylation immunoblot

ScGT1 and GT1 cells were treated for 6 days with mAbs (5 µg/mL) refreshing the medium after 3 days. Cells were then washed twice with cold PBS 1X (GIBCO/Invitrogen, Irvine, California, USA) and incubated for 10 min on ice, in lysis buffer (50 mM Tris–HCl (pH 7.4) 150 mM NaCl, 1% Triton X-100, 2 mM Na3V O4 and a mixture of protease inhibitors (Roche)). The cell extracts were then centrifuged at 2,300 × g for 5 min. The supernatant was stored at −80 °C prior to use. Total protein concentration was determined using the Bicinchoninic acid assay (Pierce). Twenty-five µg of total proteins were separated by 12% SDS-PAGE and transferred to PVDF membranes (EMD Millipore, Billerica, Massachusetts, USA). These were then blocked in 5% non-fat dried milk in TBS-T for 1 h at RT before overnight incubation at 4 °C with primary antibodies against ERK (#9107; Cell Signaling Technology, Danvers, Massachusetts, USA) or phospho-ERK (#9101; Cell Signaling Technology, Danvers, Massachusetts, USA). After 3 washes in TBS-T, the membranes were incubated for 1 h at RT in HRP-conjugated secondary antibody (1:2,000) (Invitrogen, Carlsbad, California, USA) diluted in blocking solution. The chemiluminescent signal was detected using the ECL kit (Amersham, Buckinghamshire, UK) on ECL Hypermax films (Amersham, Buckinghamshire, UK). Densitometric analysis was performed using a Molecular Imager ChemiDoc XRS System equipped with Quantity One software (Bio-Rad, Hercules, California, USA).

Thiazolyl blue tetrazolium bromide (MTT) viability assay

ScGT1 and GT1 cells were incubated in a 96-well, tissue culture-treated plate for 5 days with different mAbs at concentration 5 µg/mL, refreshing medium after 3 days. Then the medium was removed and the cells were incubated with 150 µL of MTT (Sigma, St. Louis, Missouri, USA) working solution (0.5 µg/mL of MTT in PBS) for 2 h at 37 °C. The solution was removed and formazan was solubilized by adding 150 µL of dimethyl sulfoxide (DMSO) to each well. Optical density was read at 560 nm and the background subtracted at 670 nm using the VersaMax plate reader (Molecular Device, Sunnyvale, California, USA).

Immunofluorescence assay

ScGT1 and GT1 cells were grown overnight on glass cover slips coated with poly-L-lysine (10 µg/mL) (Sigma, St. Louis, Missouri, USA) before fixation in 4% paraformaldehyde in PBS for 20 min at RT. Cells were permeabilized with 0.1% Triton X-100 in PBS for 10 min at RT and then treated for 5 min at RT with 3M Guanidine HCl (Pierce, Waltham, Massachusetts, USA) in PBS. After 3 washes in PBS, cells were blocked for 1 h at RT in 5% normal goat serum (VECTOR Laboratories, Burlingame, California, USA) in PBS. After blocking, cells were incubated at RT for 2 h with primary antibody (5 µg/mL) in the same blocking solution. Cells were washed 3 times with PBS and further incubated with secondary antibody conjugated with AlexaFluor 488 (Invitrogen, Irvine, California, USA; diluted 1:500 in blocking solution) for 1 h at RT in the dark. Cells were further washed as described above, before mounting in Vectashield with DAPI (VECTOR Laboratories, Burlingame, California, USA). Images were acquired with a DMIR2 confocal microscope equipped with Leica Confocal Software (Leica, Wetzlar, Germany).

Surface plasmon resonance (SPR)

SPR analysis was conducted using the Biacore 2000 biosensor system (GE Healthcare, Little Chalfont, UK). Recombinant MoPrP (200 RU) was immobilized on a CM5 sensor chip (GE Healthcarre, Little Chalfont, UK) using standard amine-coupling chemistry. Each mAb was flown over the bound recMoPrP at different concentrations (0, 25, 50, 100, 200 and 400 nM) in HBS-N buffer (10 mM HEPES, 150 mM NaCl), pH 7.4. The association between the antibodies with the immobilized protein was monitored for 4 min followed by 10 min of dissociation. The results were analyzed using the BIA evaluation software (GE Healthcare, Little Chalfont, UK).

Results

Monoclonal antibodies production and epitope mapping

To generate PrP-specific mAbs, three Prnp0/0 mice were immunized with recHuPrP as described in the ‘Materials and Methods’ section. Two mock injected animals served as negative controls. Mouse sera were tested in serial dilutions by indirect ELISA with recHuPrP coated on 96-well microtiter plates. The detection limit was reached at serum dilutions of 10−6 for all three animals, suggesting a strong humoral immune response with high antibody titers (Fig. S1). Mouse m#2 was chosen for splenocyte isolation and cell fusion. A large number of hybridoma cell lines were grown. Cells were selected and cloned according to the reactivity of the mAbs to recHuPrP by ELISA. The antigen used for clone selection was the same as the one used for the immunization of mice. Four cell lines producing mAbs against PrP were identified. The mAbs chosen were designated as DE10, DC2, EB8 and EF2. They were all defined as immunoglobulin subclass IgG2a. Their epitopes were analyzed by direct mapping (Fig. S2, Fig. 1) and inhibition assay (Fig. 2). Suggested epitopes were all at the N-terminal end of the prion protein between aa 26–52 (Fig. 3).

Figure 1 Epitope mapping of mAbs through direct binding.

Synthesized peptides from HuPrP (1–11) and MoPrP (12–15) were coated to the microplates at concentration 2 µg/mL. A detailed description of the peptides used in this assay is provided in Fig. S2. The four monoclonal antibodies DE10, DC2, EB8 and EF2 were added in concentration 5 µg/mL. Bars indicate binding of antibodies to peptides.

Figure 2 Epitope mapping of mAbs through competitive ELISA.

RecHuPrP (5 µg/mL) was coated to 96-well microtiter plate as antigen. Mixtures of the four monoclonal antibodies (1 µg/mL for EB8 and 0.2 µg/mL for the others) and synthesized peptides from HuPrP (1–11) and MoPrP (12–15) (0.8 µg/mL) were added. A detailed description of the peptides used in this assay is provided in Fig. S2. Bars indicate inhibition of monoclonal antibodies’ binding.

Figure 3 Suggested epitopes for the four monoclonal antibodies based on direct mapping and competitive ELISA assay.

The epitopes recognized by the four mAbs (26–34aa for EB8; 35–46aa for DC2; 44–52aa for DE10 and 47–52 for EF2) are highlighted in a multi-alignment among the amino acidic N-terminal sequences of mature PrP from the following species: human (Homo sapiens; P04156), mouse (Mus musculus; P04925), golden hamster (Mesocricetus auratus; P04273), sheep (Ovis aries; P23907), cow (Bos taurus; P10279), red deer (Cervus elaphus; P67987), rat (Rattus norvegicus; P13852) and rabbit (Oryctolagus cuniculus; Q95211).

mAbs affinity for PrP

The affinity of each mAb for recombinant mouse PrP (recMoPrP) was evaluated by surface plasmon resonance (SPR) analysis as detailed in the ‘Materials and Methods’ section. All mAbs have a high affinity for PrP, in the nM and sub-nM range (Table 1). Among them, the mAb DC2 exhibited the highest affinity with a binding affinity constant (KD) equal to 6.14 × 10−10 M. The mAb EB8 instead showed the lowest one with a KD of 1.71 × 10−8 M.

Table 1 Affinity constants (KD) of mAbs for recombinant mouse PrP (recMoPrP) as probed by surface plasmon resonance (SPR) assays.

mAb	KD (M)	
EB8	1.714 × 10−8	
DC2	6.144 × 10−10	
EF2	3.084 × 10−9	
DE10	1.071 × 10−8	

mAbs binding to PrP of different species

The specificity of the newly generated mAbs for PrP was tested by Western blot on brain homogenates from PrP-wild type and knockout mice. All four mAbs recognized the three glycosylated forms of PrPC (un-, mono- and di-glycosylated) in wild type mouse homogenates but no signal was detected in PrP-knockout samples (Fig. 4).

Figure 4 Immunoreactivity of mAbs probed by Western blot.

Homogenates from human (Hu), bovine (Bo), hamster (Ha), sheep (Sh), deer (De), rabbit (Ra), rat (Rat), mouse (Mo) and Prnp0/0 mouse (koMo) brain tissues were analyzed by Western blot using the four monoclonal antibodies. Different patterns of detection were observed. Samples from human, hamster, rat and mouse tissues were consistently detected by all the mAbs while bovine, sheep, deer and rabbit samples were poorly recognized by the EB8 antibody. Rabbit PrP was not detected by DC2 antibody. Non-contiguous lanes are highlighted (white lines).

The cross-reactivity of the four mAbs to PrP from other species was also assessed. A panel including human, bovine, hamster, sheep, deer, rabbit and rat brain homogenates was used (Fig. 4). Western blots showed similar binding patterns for all the mAbs on human, hamster and rat PrPs. On the contrary deer, sheep and bovine PrPs were recognized by all mAbs with the exception of EB8. Rabbit PrP was recognized by EF2 and DE10 but not by EB8 and DC2 mAbs (Fig. 4). The presence of an additional glycine residue within the EB8 epitope of deer, sheep, bovine and rabbit PrPs explains the lack of binding of EB8 to these proteins (Fig. 3). Instead the non-synonymous substitution of a glycine with a serine within the DC2 epitope of rabbit PrP explains the lack of binding of DC2 (Fig. 3).

Immunoreactivity of the mAbs to PrPC and PrPSc in GT1 cells

Once the specificity of the four mAbs was assessed, we tested by immunofluorescence whether they could bind PrPC and PrPScin situ. The mouse hypothalamic GT1-1 cell line, chronically infected with the RML scrapie strain, was used in the screening. Infected (ScGT1) and uninfected (GT1) cells were grown on coverslips and stained as detailed in the ‘Materials and Methods’ section. PrP localization was investigated by confocal microscopy.

The four mAbs were able to stain native PrP in situ, showing similar staining patterns. Specifically, the cell membrane and the perinuclear region were clearly immunostained in GT1 and ScGT1 cells (Figs. 5A–5B). Such PrP distribution is consistent with previous reports on GT1 cells (Marijanovic et al., 2009). Interestingly, while the perinuclear staining in ScGT1 was more homogeneous around the nucleus, in GT1 cells the staining was more concentrated in one area. At least in uninfected cells, the signal presumably derives from PrPC recycling between the cell surface and the endocytic compartment. However, under those experimental conditions we could not ascertain whether the signal coming from infected cells was comprehensive of PrPSc as well. To answer this question, cells were pre-treated with guanidinium isothiocianate for a few minutes prior to the incubation with mAbs. This chaotropic agent is widely used to denature PrPSc and unmask its buried epitopes (Yamasaki et al., 2012). In our case, no relevant changes in PrP staining were appreciated before or after guanidinium treatment (Figs. 5A–5B). Most likely, the distal epitopes recognized by the four mAbs are accessible in both PrPC and PrPSc.

Figure 5 Immunolocalization of PrP in GT1 cells.

GT1 (A) and ScGT1 (B) cells were fixed with PFA and PrP was immunostained with the four mAbs (in green) as detailed in ‘Materials and Methods’. Nuclei were counterstained with DAPI (in blue). On the right of each panel, merged images are shown. All the antibodies show a similar pattern. The cell membrane and the perinuclear region are stained. No difference was observed after guanidinium treatment. Images are representative of at least three coverslips. Scale bars, 20 µm.

The use of mAbs in histopathology

After the biochemical characterization, the mAbs were probed for their ability to stain prions in IHC experiments, the gold standard for the definitive diagnosis of human prion diseases (Budka et al., 1995). According to the targeted epitope, anti-prion mAbs exhibit different staining patterns. Interestingly, in a study comparing 10 antibodies against epitopes spanning the whole PrP sequence, N-terminus mAbs showed a weaker immunoreactivity compared to antibodies against the midregion of the protein. In addition, while the N-terminus mAbs were able to stain coarser and plaque-type PrP deposits, they stained weakly or not at all fine granular or synaptic deposits (Kovacs et al., 2002).

In our tests, cerebellar sections from healthy individuals and from a patient with sporadic CJD were stained according to standard protocols for optimal PrPSc immunodetection in tissue sections (Hegyi et al., 1997). All four mAbs were able to recognize primitive plaques derived from prion deposition in the Purkinje cell layer and in the internal granule cell layer of the sCJD patient (Fig. 6). Although IHC was performed on consequent slices of cerebellum of the same sCJD case, significant differences in the intensity of mAbs reactions can be observed, probably due to differential exposition of N-terminal epitopes upon pretreatment of the tissue samples. Interestingly, all four mAbs reacted stronger to plaques’ rims than their cores. Moreover, consistent with the aforementioned study, prion synaptic deposits were not strongly immunolabeled. This finding might reflect a higher accessibility of the flexible tail of PrP in amyloid states (plaques and plaque-like aggregates) compared to fine deposits (Nakamura et al., 2000).

Figure 6 Immunohistochemistry on human tissue samples.

Immunohistochemistry of PrPSc deposits in the cerebellum of a sCJD patient (A–D) and of a non-CJD control (E–H). Immunolabeling was performed with 5 µg/mL of mAbs EB8, DC2, DE10 or EF2, respectively. Magnified: 200 ×.

Prion replication inhibition

Several anti-PrP antibodies have shown the ability to block prion replication if added to the culture media of prion-infected cells (Peretz et al., 2001). Most inhibitory antibodies reported so far recognize epitopes in the C-terminal domain, in particular within the helix α1 (Miyamoto et al., 2005). Thus, we were particularly keen to test whether our panel of mAbs was able to inhibit prion propagation in ScGT1 cells as well. As preliminary step, we first excluded the possibility that mAbs could exert any cytotoxic effect if incubated with the cells. For this purpose, MTT cell viability assays were performed on both ScGT1 and GT1 cells. All the mAbs were tested at the concentration of 5 µg/mL for 5 days and none of them showed any statistically significant effect on cell viability (Fig. S3).

Subsequently, inhibition experiments were carried out by incubating ScGT1 cells with increasing concentrations of purified and sterile mAbs (1, 2.5, 5, 7.5 µg/mL) for 6 days, refreshing medium on the third day. The levels of PrPSc after proteinase K digestion were used as read-out for the degree of inhibition. The mAbs DE10, DC2 and EF2 promoted a complete clearance of PrPSc signal starting from 2.5 µg/mL (Fig. 7). Only EB8 was not able to inhibit prion replication completely, even at the highest concentration tested, although a dramatic decrease of PrPSc was observed at higher concentrations (Fig. 7).

Figure 7 N-terminal mAbs can inhibit prion replication.

RML infected GT1 cells were treated for 6 days with increasing concentrations (0, 1, 2.5, 5 and 7.5 µg/mL) of EB8, DE10, DC2 and EF2 mAbs, refreshing medium the third day. Cell lysates were digested with proteinase K (PK + lanes) and PrPSc levels checked by Western blot using Fab D18 for detection. As positive control (PC), cell lysates from uninfected GT1 cells were also digested. About 25 µg of total proteins were loaded as control (PK—lanes). DE10, DC2 and EF2 mAbs promoted a complete clearance of prions starting from the lowest concentration tested whilst cells treated with EB8 showed a residual signal of PrPSc even at the highest concentration of antibody. Images are representative of three independent experiments.

To assess if the prion replication inhibition was a stable or transitory phenomenon a time-course experiment was conducted. ScGT1 cells were treated for one week with the different mAbs at the concentration of 5 µg/mL. The antibodies were subsequently removed from the media and the cells were cultured for an additional month, checking the levels of PrPSc every week by PK digestion. All the antibodies showed a stable effect over time, since no appreciable come-back of PrPSc was detected in the 4 weeks after the treatment (Fig. 8).

Figure 8 Time-course analysis of mAb-induced prion clearance.

ScGT1 cells were incubated for 1 week with 5 µg/mL of mAbs. Untreated cells were used as negative control (NC). After the initial treatment, cells were split and cultured in absence of mAbs for 1 month. Cell lysates were digested with proteinase K (PK + lanes) and PrPSc was probed by Western blot using Fab D18. PrPSc levels were analyzed after one (1 w), two (2 w), three (3 w) and four (4 w) weeks after the treatment to evaluate the stability of clearance during time. Prions were not detectable in treated cells one month after the mAbs incubation. Just a slight signal from PrPSc was found in EB8 treated ScGT1 cells. Images are representative of three independent experiments. Lanes were run on the same gel but were non-contiguous (white lines).

ERK pathway analysis upon mAbs treatment

PrPC can trigger signals inside the cytosol when clustered on the cell membrane (Mouillet-Richard et al., 2000). Among the different pathways, PrPC was shown to modulate the extracellular regulated kinase (ERK) 1/2 cascade either in neuronal or non-neuronal cells (Schneider et al., 2003). Indeed, prion infection was demonstrated to aberrantly increase the levels of the ERK complex in its active form both in vitro and in vivo (Didonna & Legname, 2010; Lee et al., 2005). Thus, we tested whether mAb-treatment could not only block prion infection, but also reset the ERK pathway to healthy levels. To this purpose, both infected and uninfected GT1 cells were incubated with the four mAbs for six days at the final concentration of 5 µg/mL, and total cytosolic extracts were tested for phospho-ERK levels by Western blot (Fig. 9). Surprisingly, although the mAb treatment cleared prions, it did not revert ERK activation to the original state but, on the contrary, it further enhanced ERK phosphorylation (at least for the mAbs DE10, DC2 and EB8). Interestingly, the treatment of uninfected cells did not increase the levels of phospho-ERK and in the case of mAb DC2, ERK activation was significantly reduced.

Figure 9 Effects of mAb treatment on ERK phosphorylation in GT1 cell line.

Infected and non-infected GT1 cells were treated with the different mAbs (5 µg/mL) for 6 days. Cytosolic proteins were extracted and the levels of the phosphorylated form of ERK 1/2 complex (pERK1/2) were probed by Western blot. The total amount of ERK (totERK1/2) was determined as internal control. The treatment with DE10, DC2 and EB8 mAbs but not EF2 induces a significant increase of phospho-ERK levels in ScGT1 compared to untreated cells. The same treatment shows no effect on GT1 cells. Only the cells incubated with DC2 exhibit a significant decrease in the levels of phospho-ERK. Statistics were performed using Student’s T-test on a set of three experiments; data were normalized on the total amount of ERK. ∗P < 0.05, ∗∗P < 0.01 versus untreated controls both for infected and non-infected cells.

Discussion

Antibodies are an invaluable tool in prion biology. Since the formulation of the “protein only” hypothesis, a plethora of antibodies have been raised against different epitopes of PrPC, and they helped shed light on the structure and function of prion proteins. However, the largest part of the commercially available antibodies targets the globular and central domains of PrPC, probably as a consequence of using the proteolytic product PrP27−30 as immuno-antigen. Even in a recent systematic attempt to develop a more comprehensive panel of antibodies spanning the whole PrPC sequence, most epitopes recognized in the N-terminal domain were located within the octapeptide repeats, with the exclusion of the very distal portion (Polymenidou et al., 2008).

Here, we report the production and the exhaustive characterization of four novel monoclonal antibodies which recognize three epitopes in the first 50 amino acids of the PrPC mature sequence, designated DE10, DC2, EB8 and EF2. The mAbs were raised against recombinant human PrP (recHuPrP 23-231) in PrPC-deficient mice and recognize PrP from different species. It is curious how only N-terminal mAbs were obtained, although mice were immunized with full-length PrP and also the selection of cell lines was performed with the same antigene. As Prnp0/0 B cells were fused to Prnp+/+ NS1 myeloma cells, one possible explanation could be that Abs to other PrP epitopes might have elicited apoptotic signals upon binding to PrPC on hybridoma cells. Thus, only non-toxic mAbs could have been spontaneously selected.

Several lines of evidence suggest that the three epitopes recognized by our panel of mAbs are linear and continuous. First, peptide array analysis univocally mapped the three epitopes in the distant N-terminus of PrP without any signal coming from other portions of the protein. Second, the mAbs are able to detect PrP in western blot assays performed in denaturing conditions. Lastly, in situ binding to PrP on the cell surface survived the treatment with guanidinium—a strong chaotropic agent. However, a recent work has highlighted how some discontinuous epitopes in PrP can remain functional following denaturing treatments due to a quick refolding in a structure still amenable to be recognized by conformation-dependent antibodies (Kang et al., 2012). Thus, the existence of conformational epitopes cannot be formally excluded. To systematically address this question, further studies employing the latest high-resolution mapping technologies will be required. For instance, mutational scanning by cell–surface display and single-molecule real-time (SMRT) deep sequencing have been recently used to identify discontinuous residues critical to ligand binding for four anti-PrP antibodies (Doolan & Colby, 2015).

Interestingly, all mAbs belong to immunoglobulin subclass IgG2a. The production of Ig2a antibodies is one of the characteristics of the Th1 type immune response (Mosmann et al., 1986). It has been shown that prion conformation significantly influences the type of immune response in Prnp0/0 mice (Khalili-Shirazi et al., 2005). We could speculate that the structure acquired by recombinant HuPrP in our immunization experiments might have shifted the immune response toward a Th1 type.

Besides their intuitive usage in diagnostics and basic research for investigating the role of the N-terminus in prion protein physiology, we were also interested in possible therapeutic applications to treat prion disorders. To date, hundreds of chemical compounds have been found to stop prion replication in vitro or in cell cultures (Sim & Caughey, 2009) but very little success was achieved in translating their properties to in vivo models, mainly due to their toxicity or inability to cross the blood brain barrier (Chang et al., 2012).

Antibodies are one of the most promising tools in developing effective cures for prion diseases (Rovis & Legname, 2014). Indeed, they have been shown to clear prion infectivity in cellular models of prion replication (Peretz et al., 2001) and to significantly delay the disease development in mice if mAbs are administered shortly after infection (White et al., 2003).

Several mechanisms have been proposed to explain antibody-mediated prion replication inhibition. Anti-PrP antibodies may slow down the conversion process by preventing interaction between PrPC and PrPSc, as the latter is believed to act as a template to refold PrPC into new molecules of the pathological conformer (Peretz et al., 2001). Alternatively, antibodies may indirectly affect prion conversion through perturbation of PrPC cellular trafficking (Feraudet et al., 2005). Finally, it has been suggested that anti-PrP antibodies can block PrPSc replication by accelerating PrPC degradation (Perrier et al., 2004).

In our inhibition experiments, three of the four mAbs (DC2, DE10 and EF2)—whose epitopes span the aa35–aa52 region—were able to lower prion conversion below the detection limit of Western blot at nM concentrations. Moreover, the inhibition resulted stable over time as pulse-chase experiments with a one-month follow-up did not show any increase in PrPSc levels after incubation with the antibodies. Three regions within the PrPC molecule (aa23–aa33, aa98–aa110 and aa136–aa158) have been shown to tightly bind PrPSc and mediate prion conversion (Solforosi et al., 2007). The epitopes recognized by our mAbs are adjacent to the region aa23–aa33, which corresponds to the polybasic domain. This may explain their high efficiency in blocking prion replication. Moreover, a novel PrPSc-specific epitope has been recently reported in the region aa31–aa47 by using mAbs raised against intact PrPSc complexes (Masujin et al., 2013).

Surprisingly, in our panel, mAb EB8 failed to promote prion clearance as efficiently as the other antibodies. Since its epitope (aa26–aa34) almost overlaps with the polybasic region we expected better performances, as the domain is one of the three replicative interfaces and it was found playing a critical role in prion conversion in vivo (Solforosi et al., 2007; Turnbaugh et al., 2012). This observation could be ascribed to the lower affinity of EB8 compared to the other mAbs as highlighted by SPR assays. In addition, it should be mentioned that an independent antibody recognizing the same epitope in ovine PrP showed results comparable to EB8 in a large inhibition screening on 145 antibodies (Feraudet et al., 2005).

Little is known about how prions mediate toxic signals to the cell. Prion infection was demonstrated to aberrantly increase the levels of the ERK complex in its active form both in vitro and in vivo (Didonna & Legname, 2010; Lee et al., 2005). We have previously demonstrated that the Fab fragment D18 does not revert ERK activation although it is very efficient in clearing prions (Didonna & Legname, 2010). D18 epitope spans from residues aa132–aa156 in the globular domain of PrP. Thus, we asked if antibodies targeting the N-terminal domain were better at decreasing the levels of phospho-ERK in ScGT1 cells. Unfortunately, our data show that ERK activation was even enhanced upon addition of mAbs to infected GT1 cells. This finding suggests that the targeted epitope is irrelevant as prion infection might have irremediably altered cell physiology. Alternatively, we could speculate that antibody treatment does not fully block prion replication but resets the process at levels below the limit of detection.

Conclusions

In summary, by combining a variety of techniques we have described a new panel of antibodies that will be useful in both basic research and diagnostics. Moreover, we have identified three of them as promising candidates for immunotherapy of prion diseases. Future studies will aim to assess their safety upon in vivo administration. Since antibodies cannot cross the blood brain barrier, the next step will be converting the mAbs into single-chain variable fragments (scFvs). scFvs are monovalent mini-antibodies maintaining the same antigen specificity of mAbs that can be easily engineered to be expressed by adeno-associate viral (AAV) vectors for intracerebral delivery (Campana et al., 2009). Indeed, a similar approach using AAV serotypes 2 and 9 to deliver several anti-PrP scFvs delayed the onset of prion pathogenesis in mice without fully blocking it (Moda et al., 2012; Wuertzer et al., 2008). Alternatively, scFvs can be fused to a cell-penetrating peptide, which can cross the blood–brain barrier and deliver the mini-antibodies to the site of action (Skrlj et al., 2013). It will be important to assess if scFvs against N-terminal epitopes are more effective in stopping prion replication compared to those targeting the globular domain.

In order to encourage the use of these novel mAbs through collaborative or independent projects, they are available for the prion research community in the cell bank at the Blood Transfusion Center of Slovenia (http://www.ztm.si/en/).

Supplemental Information

Figure S1 Immune response analysis

Immune sera from three mice and two negative controls were probed by ELISA against recombinant human PrP (recHuPrP). Serial dilutions were prepared for each sample. The sera from immunized mice were able to recognized recHuPrP up to the 10−6 dilution while both negative controls gave no reaction.

Click here for additional data file.

Figure S2 Peptides used for epitope mapping

Panel of the synthesized peptides from HuPrP sequence (1–11) and MoPrP sequence (12–15) used for epitope mapping.

Click here for additional data file.

Figure S3 Cell viability of mAb-treated cells

Both GT1 and ScGT1 cells were treated with the different mAbs for 5 days, refreshing the medium on the third day. Then cell viability was evaluated by MTT assay according to the procedure described in the ‘Materials and Methods’ section. No statistical differences in term of cell viability were found in mAb-treated cells compared to untreated controls. For every mAb the average values from 5 wells are expressed as percentages of cell viability referred to untreated cells.

Click here for additional data file.

Supplemental Information 4 Raw data

Raw data for immunoblots, ELISA assays, MTT assays, immunofluorescence and Surface Plasmon Resonance (SPR) experiments.

Click here for additional data file.

Additional Information and Declarations

Competing Interests

Author Contributions

Human Ethics

Animal Ethics

The authors declare there are no competing interests.

Alessandro Didonna and Anja Colja Venturini conceived and designed the experiments, performed the experiments, analyzed the data, wrote the paper, prepared figures and/or tables, reviewed drafts of the paper.

Katrina Hartman and Tanja Vranac performed the experiments, analyzed the data, wrote the paper, prepared figures and/or tables, reviewed drafts of the paper.

Vladka Čurin Šerbec and Giuseppe Legname conceived and designed the experiments, analyzed the data, contributed reagents/materials/analysis tools, wrote the paper, reviewed drafts of the paper.

The following information was supplied relating to ethical approvals (i.e., approving body and any reference numbers):

Approval for research involving human material has been obtained from the Slovenian National Medical Ethics Committee with decision dated January 15, 2008. Post mortem brain tissue of a patient who was clinically suspected for CJD was analyzed by immunohistochemistry without the patient’s consent because such analysis is obligatory by a ministerial decree in the purpose of TSE surveillance (Official Gazette of the Republic of Slovenia, 2/2001). Human brain samples for immunohistochemistry were obtained from the Institute of Pathology, Faculty of Medicine, University of Ljubljana, Slovenia.

The following information was supplied relating to ethical approvals (i.e., approving body and any reference numbers):

All experiments involving animals were performed in accordance with European regulations [European Community Council Directive, November 24, 1986 (86/609/EEC)]. Experimental procedures were notified to and approved by the Italian Ministry of Health, Directorate General for Animal Health (notification of 17 Sept. 2012). All experiments were approved by the local authority veterinary service and by SISSA Ethics Committee. All reasonable efforts were made to ameliorate suffering. All mice were obtained from the European Mutant Mouse Archive.

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
