# Peer review of "Characterization of four new monoclonal antibodies against the distal N-terminal region of PrPc"

_PeerJ, doi:10.7717/peerj.811_

## Round 0.1 · original submission · Minor Revisions

Please revise your manuscript according to the suggestions expressed by Reviewer 2 and adding detailed information on the availability of the hybridomas

·

Basic reporting

The authors describe the generation and characterization of four novel anti-PrP antibodies, directed against the N-terminal, unstructured region of the protein. Interestingly, these antibodies show anti-prion activity in cell-based assays. The manuscript is well written, the results presented clearly, and the main findings discussed adequately. This reviewer found no issues with the study. Perhaps the only minor point could be a more accurate and extensive citation of the literature. The study seems to adhere to the journal's policy.

Experimental design

The study describes original research. The experiments are well designed and appear solid. The research seems to have been carry out in conformity to relevant ethical standards.

Validity of the findings

The results are clear and largely consistent with previous literature. The conclusions derived from the data are appropriate or reasonable.

Additional comments

Well-designed study reporting novel and interesting findings about four novel anti-PrP antibodies. The biochemical characterization of the antibodies is extensive, as well as the definition of their biological properties.

Reviewer 2 ·

Basic reporting

The submitted article appears to conform to journal standards.

Experimental design

This paper describes the isolation and characterization of four novel antibodies with specificity for the prion protein. The techniques used to isolate the antibodies are well-established and appropriate and the characterization is thorough.

Validity of the findings

The findings appear valid and are for the most part clearly explained.

Additional comments

A significant focus of the current manuscript deals with the identification of the new antibodies' epitopes. Recent publications by the Telling group (JBC 2013) and Colby group (JMB 2014) describe advanced methods for epitope determination and the conformational basis thereof. These earlier works should be discussed in the context of the current manuscript, especially with regards to the discussion of whether discontinuous or linear epitopes are present in the antibodies described here.

Although it is generally accepted scientific practice to make propagateable reagents described in research articles available to the scientific community, the prion disease research community has a poor track record in this regard. It would be refreshing to see the hybridomas deposited with an antibody cell bank, such as that at the University of Iowa.

The legend to figures 1 and 2 could be made more clear by adding a description of the peptides used for epitope mapping.

---

## Round 0.2 · accepted · Accept

The manuscript is accepted for publication